# Hydroxyl-rich macromolecules enable the bio-inspired synthesis of single crystal nanocomposites

Yi-Yeoun Kim [1]*, Robert Darkins[2], Alexander Broad[2], Alexander N. Kulak[1], Mark A. Holden[1], Ouassef Nahi[1], Steven P. Armes[3], Chiu C. Tang[4], Rebecca F. Thompson[5], Frederic Marin[6], Dorothy M. Duffy [2]* & Fiona C. Meldrum[1]*

Acidic macromolecules are traditionally considered key to calcium carbonate biomineralisation and have long been first choice in the bio-inspired synthesis of crystalline materials. Here, we challenge this view and demonstrate that low-charge macromolecules can vastly outperform their acidic counterparts in the synthesis of nanocomposites. Using gold nanoparticles functionalised with low charge, hydroxyl-rich proteins and homopolymers as growth additives, we show that extremely high concentrations of nanoparticles can be incorporated within calcite single crystals, while maintaining the continuity of the lattice and the original rhombohedral morphologies of the crystals. The nanoparticles are perfectly dispersed within the host crystal and at high concentrations are so closely apposed that they exhibit plasmon coupling and induce an unexpected contraction of the crystal lattice. The versatility of this strategy is then demonstrated by extension to alternative host crystals. This simple and scalable occlusion approach opens the door to a novel class of single crystal nanocomposites.

[1] School of Chemistry, University of Leeds, Woodhouse Lane, Leeds LS2 9JT, UK. [2] Department of Physics and Astronomy, University College London, Gower Street, London WC1E 6BT, UK. [3] Department of Chemistry, University of Sheffield, Brook Hill, Sheffield S3 7HF, UK. [4] Diamond Light Source, Harwell Science and Innovation Campus, Didcot, Oxfordshire OX11 0DE, UK. [5] The Astbury Biostructure Laboratory, Astbury Centre for Structural Molecular Biology, Faculty of Biological Sciences, University of Leeds, Leeds, UK. [6] UMR CNRS 6282 Biogeosciences, Université de Bourgogne–Franche-Comté, 6 Boulevard Gabriel, 21000 Dijon, France. *email: y.y.kim@leeds.ac.uk; d.duffy@ucl.ac.uk; F.Meldrum@leeds.ac.uk

Biological systems achieve exquisite control over inorganic materials synthesis, generating biominerals whose properties are optimised for their functions[1]. Significant efforts have therefore been made to identify the strategies used to control mineralisation, with the ultimate goal of creating superior materials synthetically[2,3]. While such control is achieved by many routes, all are united by one common feature; nature directs mineralisation using organic molecules. In calcium carbonate systems these are conventionally subdivided into an insoluble fraction comprising heavily crossed-linked macromolecules and a soluble matrix[4]. Early biochemical investigations of the latter highlighted the highly acidic macromolecules[5–7], where these had striking effects on crystal morphologies[8], and their incorporation within crystals gave rise to changes in microstructure[9,10]. This led to the development of one of the basic tenets of biomineralisation; that organisms employ highly acidic macromolecules to control CaCO₃ crystallisation[1].

Bio-inspired crystallisation methods have therefore principally focused on acidic molecules, where these can be highly effective in controlling morphologies[3,11,12] orientations[13], and polymorph[14]. Occlusion of small amounts of acidic macromolecules also enhanced the mechanical properties of calcite crystals[15,16]. In contrast, most basic and low-charge molecules induce only minor changes in the appearance of the product crystals[17]. However, increasingly sophisticated methods for identifying the macromolecules involved in CaCO₃ biomineralisation suggest a more complex picture in which regulation is achieved using an array of molecules including glycosylated proteins, peptides, metabolites and polysaccharides[18]. Further, binding assays have shown that weakly acidic peptides can associate strongly with CaCO₃[19–21], and atomic force microscopy (AFM) and molecular dynamics (MD) simulations have demonstrated the strong binding of small hydroxyl-functionalised molecules[22,23]. This suggests potential roles for low-charge and basic molecules in controlling crystal properties.

Here, we use gold nanoparticles (NPs) functionalised with proteins and polymers to investigate the influence of low-charge additives on calcite crystals, including their incorporation within the latice. The latter is a key biomineralisation strategy. However, it is generally over-looked in synthetic systems unless the incorporated species is coloured or sufficiently large to be imaged. We demonstrate that low-charge, hydroxyl-rich polymers and glycoproteins show outstanding activities, driving the formation of deep red, single crystals containing up to 37 wt% NPs. This is achieved in the absence of any signature change in morphology and runs counter to the expectation that crystallisation usually excludes impurities. AFM, high-resolution powder X-ray diffraction (HR-PXRD), and MD simulations provide insight into the activities of these macromolecules and their influence on the crystal lattice, and we show that our approach can be extended to alternative crystals. This work therefore offers a powerful and versatile strategy for creating nanocomposites with new structures and properties.

## Results

**Calcite crystallisation with protein-functionalised gold NPs.** CaCO₃ was precipitated in the presence of 14 nm gold NPs functionalised with coronas of two commercially available proteins (Protein-NPs): alpha-1-acid glycoprotein (GP) and bovine serum albumin (BSA) (Fig. 1). GP comprises very high levels of carbohydrates (45%) of which 11% are sialic acid groups, while BSA is non-glycosylated and comprises 17% acidic and 14% basic residues and 10 sites of phosphorylation. Both are somewhat acidic by isoelectric point (PI) (Supplementary Table 1). These Protein-NPs exhibited hydrodynamic diameters of 31–36 nm,

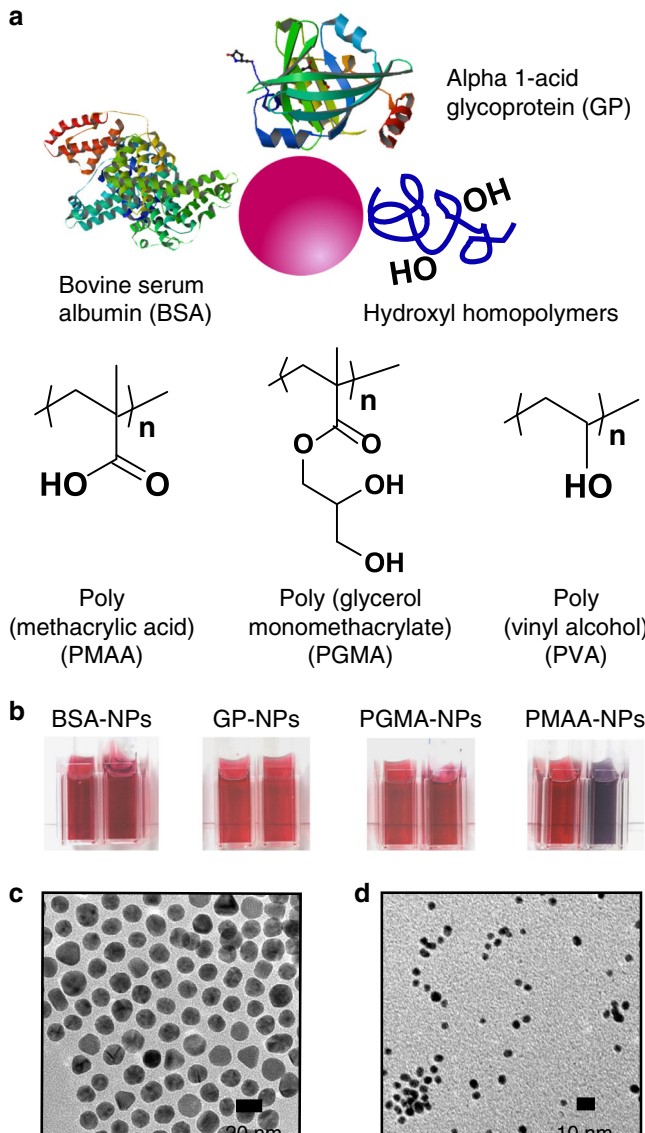

**Fig. 1** Characterisation of functionalised gold nanoparticles. **a** Schematic diagram and chemical structures summarising the different macromolecules used in the study. Protein structures of alpha-1-acid glycoprotein (GP) and BSA are PDB(3KQ0) and PDB(3V03), respectively. **b** Photographs of solutions of functionalised nanoparticles in the presence of [Ca²⁺] = 0 mM (left) and [Ca²⁺] = 20 mM, pH 9 (right). TEM images of **c** 14 nm and **d** 4 nm diameter gold nanoparticles coated with alpha-1-glycoprotein.

comprised 40–50 wt% organics, and possessed zeta potentials of −4 to −7 mV at pH 9 and [Ca²⁺] = 20 mM (Supplementary Fig. 1, Supplementary Table 2). They also both possess excellent colloidal stability under the solution conditions employed (Fig. 1b).

Precipitation of calcium carbonate in the presence of 0.05 wt% Protein-NPs at [Ca²⁺] = 20 mM using the ammonium carbonate diffusion method[24] yielded deep red/black calcite crystals, consistent with high levels of NP incorporation (Figs. 2a and 3a). Other than their colour, these crystals were identical in shape to calcite crystals precipitated in the presence of the proteins alone (Figs. 2b and 3b). The GP-NPs gave superior occlusion (≈13 wt% GP-NPs as compared with 11 wt% BSA-NPs) as determined by atomic absorption spectroscopy (AAS). Examination of cross-sections through crystals containing GP-NPs

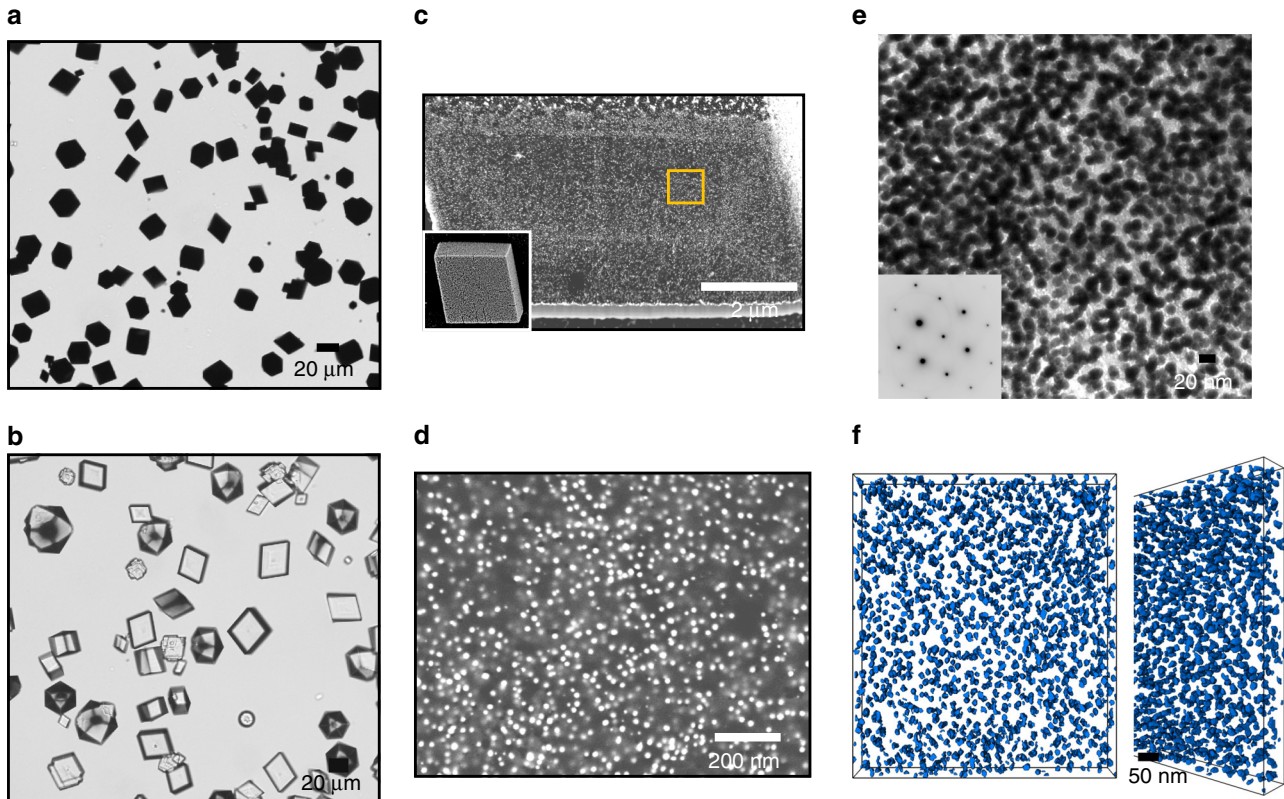

**Fig. 2** Calcite crystals precipitated in the presence of GP and 14 nm GP-NPs. **a, b** Optical micrographs of crystals precipitated at $[Ca^{2+}] = 20$ mM in the presence of **a** 0.05 wt% GP-functionalised 14 nm NPs and **b** 0.1 wt% of GP only. **c, d** SEM images of an FIB cross-section of calcite crystals containing GP-NPs. **d** is a high-magnification image of **c**. **e** TEM bright field image and SAED pattern of a crystal, showing that the host calcite is a single crystal. **f** Cryo-electron tomogram of a crystal containing 13 wt% Au showing the uniform distribution of the NPs within the host crystal.

(Fig. 2c–f) and BSA-NPs (Fig. 3c) demonstrated that the NPs were well-distributed throughout the entire crystal in both cases. This was further supported by transmission electron microscopy (TEM) and cryo-electron tomography of a focussed ion beam (FIB)-cut lamella from a crystal containing GP-NPs, where selected area electron diffraction (SAED) confirmed single crystallinity (Fig. 2e). The images also confirmed the absence of aggregates and revealed particle separations of ≈30 nm (Fig. 3f, Supplementary Video 1). Surprisingly, both Protein-NPs had little effect on crystal morphologies despite such effective occlusion.

Although neither GP nor BSA are considered highly acidic, both contain some acidic groups that may influence $CaCO_3$ precipitation. Further insights into their activities were therefore obtained by chemically reducing the acidity by (i) amination of the carboxylate groups on BSA and (ii) removal of the terminal sialic acid groups on the polysaccharide side chains of GP. The zeta potentials of these protein-NPs at pH 9 and $[Ca^{2+}] = 20$ mM increased from –7 to –4 and –6 to 6 mV, respectively, following these treatments. Both of these modified Protein-NPs were still effectively occluded (9 wt% for aminated BSA-NPs and 11 wt% for desialylated GP-NPs; Supplementary Table 3), indicating that the acidity of the proteins is not the principal factor driving their occlusion (Supplementary Figs. 2 and 3).

**Calcite crystallisation with hydroxyl polymer-functionalised gold NPs.** These results demonstrate that low-charge proteins can be highly effective at mediating NP incorporation. This principle was then translated to a purely synthetic system by functionalising the gold NPs with hydroxylated homopolymers (Polymer-NPs): poly(2-hydroxyethyl methacrylate) (PHEMA), poly

(glycerol monomethacrylate) (PGMA), and poly(vinyl alcohol) (PVA). These Polymer-NPs had comparable sizes and lower zeta potentials (−2 to 0 mV) than the Protein-NPs (Supplementary Table 2). PHEMA and PGMA have side chains terminated by single hydroxyl and diol groups, respectively, while the hydroxyl groups are directly bound to the polymer backbone in PVA. Notably, the diol functionality of PGMA resembles a key structural motif of the glycoproteins. Calcium carbonate was precipitated in the presence of 0.05 wt% 14 nm Polymer-NPs using the ammonium carbonate diffusion method unless otherwise stated.

PHEMA is poorly soluble in water, which limits its ability to coat gold NPs. However, crystals grown in the presence of PHEMA-NPs were pale pink, demonstrating some occlusion (Supplementary Fig. 4). PGMA-NPs, in contrast, were occluded at high levels, estimated as 33 wt% NPs by AAS and 34 wt% by PXRD (Fig. 4h). Scanning electron microscopy (SEM) and scanning transmission electron microscopy (STEM) HAADF images of cross-sections show the high density of NPs (Fig. 4c, d), while serial FIB sections and STEM tomography images confirm their uniform distribution (Fig. 4e, Supplementary Videos 2 and 3). Analysis of the tomography images gives average NP separations of 20 nm (Supplementary Note 1) and a loading of 41 wt%, where electron tomography of nanocomposites may overestimate the fraction of the particulate phase due to image dilation along the z-direction[25]. Importantly, the calcite matrix remained single crystal despite such high NP loadings, as conclusively shown by SAED and single crystal X-ray diffractometry (Fig. 4f, g). Comparison with pure calcite shows that these 14 nm particles cause a lattice expansion along both the $a$ and $c$ axes ($\Delta a = 0.00112(2)$ and $\Delta c = 0.00101(2)$). Finally, PVA-NPs

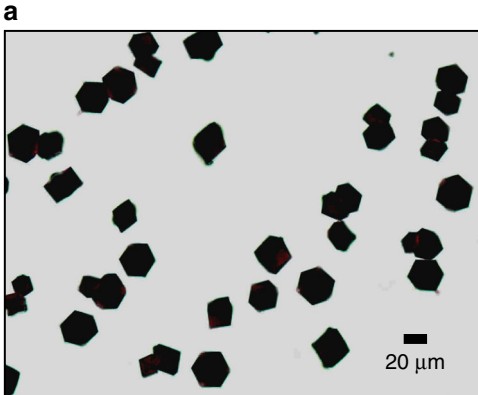

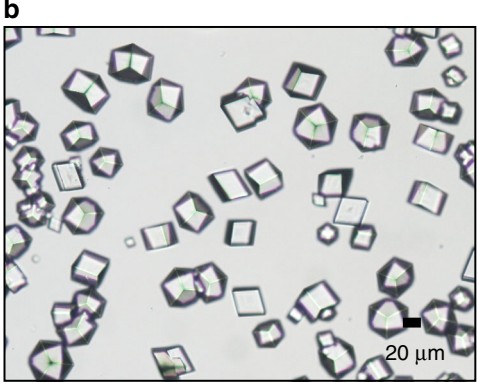

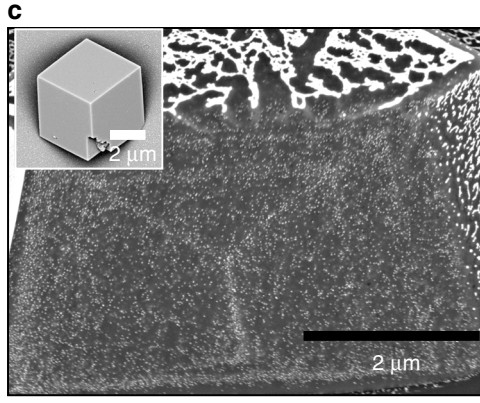

**Fig. 3** Calcite crystals precipitated in the presence of BSA and 14 nm BSA-NPs. **a, b** Optical micrographs of crystals precipitated at $[Ca^{2+}] = 20$ mM in the presence of **a** 0.05 wt% BSA-functionalised 14 nm NPs and **b** 0.1 wt% of BSA only. **c** SEM images of an FIB cross-section of calcite crystals containing BSA-NPs.

were also explored, where PVA is cheap and commercially available. This simple polymer led to the formation of pink crystals consistent with moderate occlusion (8 wt%) (Fig. 5a, b).

The behaviour of the PGMA-NPs was then compared to NPs functionalised with poly(methacrylic acid) (PMAA), where additives with carboxylic acid groups are effective growth modifiers for $CaCO_3$[11]. The PMAA-NPs exhibited zeta potentials of −40 mV at $[Ca^{2+}] = 1$ mM and pH 9 and were only partially stable in a solution of $[Ca^{2+}] = 2$ mM (Fig. 1b). The PGMA-NPs, in contrast, were still stable at $[Ca^{2+}] = 500$ mM, which opens the door to the production of large quantities of these nanocomposites. The performances of the PMAA-NPs and PGMA-NPs were therefore compared at $[Ca^{2+}] = 1.5$ mM and low particle concentrations (0.01 wt%) (Supplementary Fig. 5). Crystals

precipitated in the presence of the PMAA-NPs were elongated and pink in colour, while those formed in the presence of the PGMA-NPs were perfect rhombohedra and paler in colour. Sections through the crystals suggested somewhat lower occlusion of the PGMA-NPs under this condition of lower Ca concentration. We also explored NPs functionalised with a basic macromolecule, poly(allylamine hydrochloride) (PAH). In common with the PMAA-NPs, PAH-NPs aggregated in the crystallisation solution[17], and either generated polycrystalline particles, or were very poorly incorporated when employed at concentrations that yielded single crystals (Supplementary Fig. 6).

**Increase in packing density**. Investigation of different solution conditions and both 4 and 14 nm PGMA-NPs showed that while a higher wt% occlusion can be achieved with larger NPs, greater number densities—and thus smaller inter-particle separations—can be achieved with smaller NPs. For example, calcite crystals precipitated using the Kitano method in the presence of 0.001 wt% 4 nm PGMA-NPs contained 2.7 wt% NPs, and showed unmodified morphologies (Fig. 7). Increase in the concentration of NPs to 0.005 wt% yielded slightly truncated rhombohedral calcite crystals containing 12 wt% PGMA-NPs. Precipitation in the presence of 0.02 wt% 4 nm PGMA-NPs led to exceptional levels of occlusion of 37 wt% (44 wt% as estimated from STEM tomography) and average particle separations of just 4–5 nm (Fig. 6a–c). A full tilt series of STEM tomography is shown in Supplementary Video 4, but the density of NPs was too high to enable tomographic reconstruction. High-resolution TEM imaging and SAED at different points across a crystal confirmed its perfection despite the high density of NPs (Figs. 6c and 7a–d). Interestingly, these high occlusion levels were also accompanied by significant changes in the crystal morphologies.

These crystals also showed an interesting optical property—plasmon coupling—where this was apparent from a significant red shift in the plasmon band from 525 nm in bulk solution to 577 nm in the nanocomposites (Fig. 6d). This shift is in good agreement with theoretical estimates (Supplementary Note 2). By comparison, aggregated NPs in solution exhibit a plasmon absorption at 535 nm, and a value of 555 nm is predicted for Au NPs embedded in calcite (refractive index = 1.66).

**Microstructure**. Selected crystals were also analysed using synchrotron HR-PXRD to determine the influence of the NPs on the crystal lattice[10,15], where changes in the peak positions and shapes are both indicative of additive occlusion, and can provide insight into the crystal/additive interactions[10,26,27]. Control samples comprising calcite containing PGMA, GP, and BSA alone all exhibited peak broadening but little peak shift, which demonstrates occlusion of these species (Fig. 8a, e, Supplementary Fig. 7). On incorporation of ≈12 wt% 4 nm PGMA-NPs (the crystals are shown in Fig. 7f) the {104}, {110}, and {006} reflections were all broadened and the {104} and {006} were shifted to larger d-spacing (corresponding to a lattice expansion) (Fig. 8b–e). Notably, large coherence lengths that were only slightly smaller than those of pure calcite were recorded, as is consistent with the assignment of these nanocomposites as single crystals. Crystals containing high loadings of 4 nm PGMA-NPs (37 wt%) gave surprising results, however, where they exhibited a lattice contraction along the a-axis. A small lattice expansion was observed along the c-axis. The overall unit cell volume was therefore *smaller* than that of pure calcite. Again, large coherence lengths were measured (Fig. 8e).

These effects can be rationalised by considering the structures of these nanocomposites. At low levels of occlusion the crystals comprise unstrained calcite and calcite that is influenced by the

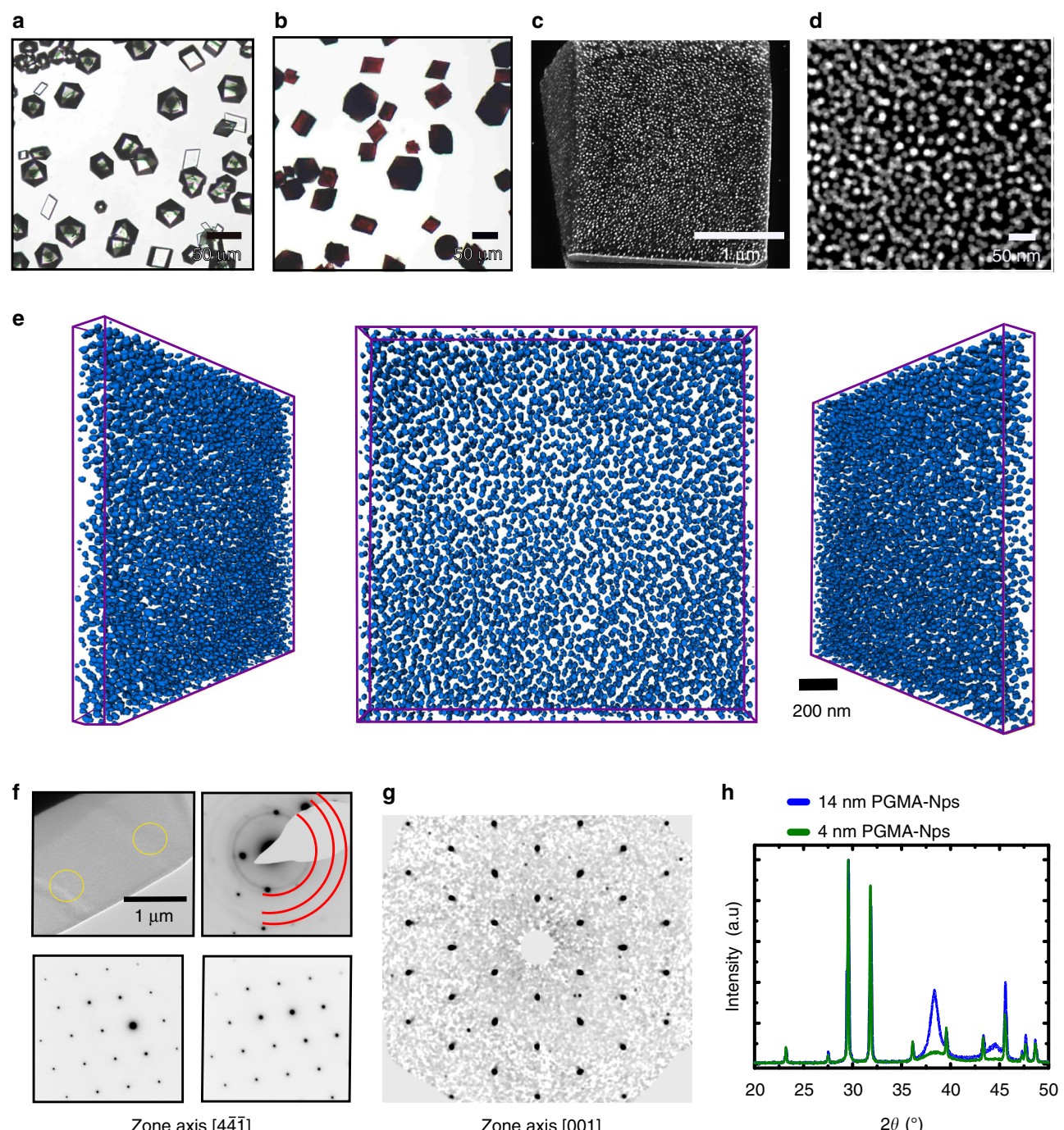

**Fig. 4** Calcite crystals precipitated in the presence of PGMA and 14 nm PGMA-NPs. **a** Optical micrograph of crystals grown in the presence of 1 wt% PGMA polymer only. **b**–**h** Analyses of crystals precipitated in the presence of 0.05 wt% of PGMA-NPs. **b** Optical micrograph, **c** SEM image of FIB cross-section and **d** STEM HAADF image of a lamellar cut from a crystal. **e** Segmented STEM tomograms of a sectioned lamella from a representative crystal shown perpendicular to the lamella and at two angles. **f** A low-magnification TEM image of a sectioned lamella from a crystal and the corresponding selected area electron diffraction (SAED) pattern, recorded off the zone axis, to enable diffraction rings from the gold NPs to be readily observed. Two further SAED patterns from the areas indicated are also shown, demonstrating the single crystal structure. **g** Single crystal diffraction pattern (recorded with a single crystal diffractometer) taken from a crystal ≈70 μm in size. The reconstructed (unwarped) layer of the diffraction pattern along the [0 0 1] direction is shown, where this was obtained by full rotations of the crystal. **h** Lab powder XRD diffractograms of PGMA-NP/calcite nanocomposite crystals containing 4 and 14 nm NPs.

occlusions. The presence of organics is associated with an increase in the lattice spacings, where this is greatest along the $c$-axis due to the elastic anisotropy of calcite[15]. Calcite crystals containing low concentrations of stabilised NPs therefore exhibit peak broadening and lattice expansion[10,28]. At very high levels of

occlusion a zone arising from the interface between the NP and crystal matrix becomes significant. Its behaviour can be considered based on the mechanics of the interface between a NP and the surrounding crystal. This was explored by performing MD simulations of the lattice distortions resulting from the

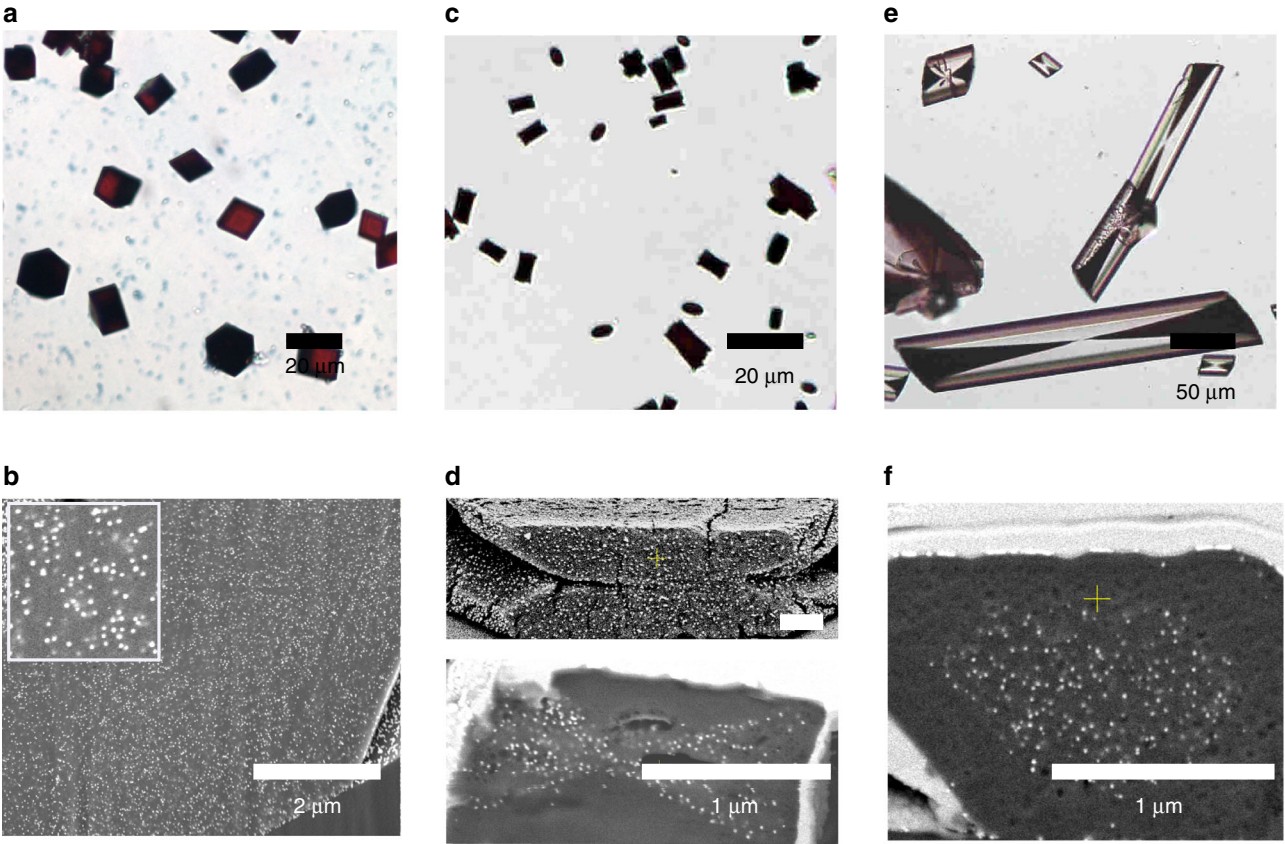

**Fig. 5** Calcite, gypsum and COM crystals precipitated in the presence of hydroxyl-NPs. **a, b** Calcite crystals precipitated in the presence of 0.05 wt% 14 nm PVA-NPs at $[Ca^{2+}] = 20$ mM. **c, d** Calcium oxalate monohydrate (COM) crystals precipitated at $[Ca^{2+}] = 0.75$ mM and $[Ox] = 0.75$ mM in the presence of 0.01 wt% 14 nm PGMA-NPs. **e, f** Calcium sulfate dihydrate (gypsum) crystals precipitated at $[Ca^{2+}] = 100$ mM and $[SO_4^{2-}] = 200$ mM in the presence of 0.08 wt% 14 nm PGMA-NPs. **a, c, e** show optical micrographs while **b, d, f** are SEM images of FIB cross-sections of the crystals.

presence of periodic arrays of 'holes' in calcite. This simple model does not consider the specific chemical/physical interactions between the occluded particles and the host lattice, but is sufficient to give insight into the origins of the observed changes in the macrostrains. Since the shape of the void occupied by the NPs is unknown, two extreme cases were considered: spherical holes and rhombohedral holes with {104} faces. The holes were ≈4 nm in diameter in an 8 nm supercell. Upon relaxation, the hydrostatic stress distributions and average lattice parameters were calculated.

In both cases, there is a positive surface stress, which induces a compressive stress in the surrounding crystal, causing a local contraction. This effect is enhanced with decreasing NP size. The spherical hole has a high surface stress and causes a decrease in lattice parameters in both directions (Fig. 9). In contrast, the rhombohedral hole has lower surface stress on the flat surfaces and at the obtuse edges, but a high tensile stress at the acute edges. This stress distribution induces a contraction in the *a*-direction and an expansion in the *c*-direction, and calculation of the lattice parameters yields values comparable to those obtained experimentally (Fig. 9). This rhombohedral hole is thermodynamically favoured and is therefore expected to be more representative of the experiments than the sphere.

**Mechanism of occlusion.** Changes in macroscopic crystal morphologies are often accompanied by additive occlusion, where both effects have the same origin—additive binding to the crystal surface. Additives can bind to calcite on the terraces, step edges and kink sites in a dynamic process that is governed by the rates

of attachment and detachment. Morphological changes occur when the soluble additives inhibit the motion of the kink sites[29], changing the rate of step propagation and causing step bunching. Occlusion occurs when the crystal grows around an additive, pinning it to the surface, and ultimately engulfing it. This can take place with limited kink site blocking and an associated morphology change[29]. As shown in our experiments, it is therefore possible to identify conditions that range from low occlusion with marked morphology changes (PMAA-NPs at $[Ca^{2+}] = 1.5$ mM) to effective occlusion with minor changes in morphology (PGMA-NPs at $[Ca^{2+}] = 20$ mM).

The mechanism of occlusion of PGMA-NPs in calcite was investigated using in situ AFM, where growing crystals were analysed in a liquid cell under slow flow of a calcium carbonate solution containing PGMA-NPs (Fig. 10a, b)[30]. Calcite grows via a step growth mechanism under the reaction conditions selected, where the steps originate from screw dislocations present on the {104} faces. Due to the symmetry of the calcite lattice, the step edges lie either acute or obtuse to the crystal faces, and additives can bind preferentially to one set of step edges[31,32].

The NPs readily adsorbed to the crystal surface when they were introduced into the liquid cell. Experiments were performed using Peak Force Tapping mode to reduce the tip-sample interactions as far as possible, while still resolving the growing crystal surface; the NPs are otherwise readily displaced by the AFM tip during scanning. (Supplementary Fig. 8) Notably, the PGMA-NPs preferentially bound to the acute steps under the slow growth conditions employed in AFM (with a ratio of ≈ 80/20), and were subsequently incorporated with minimal changes to the shape of

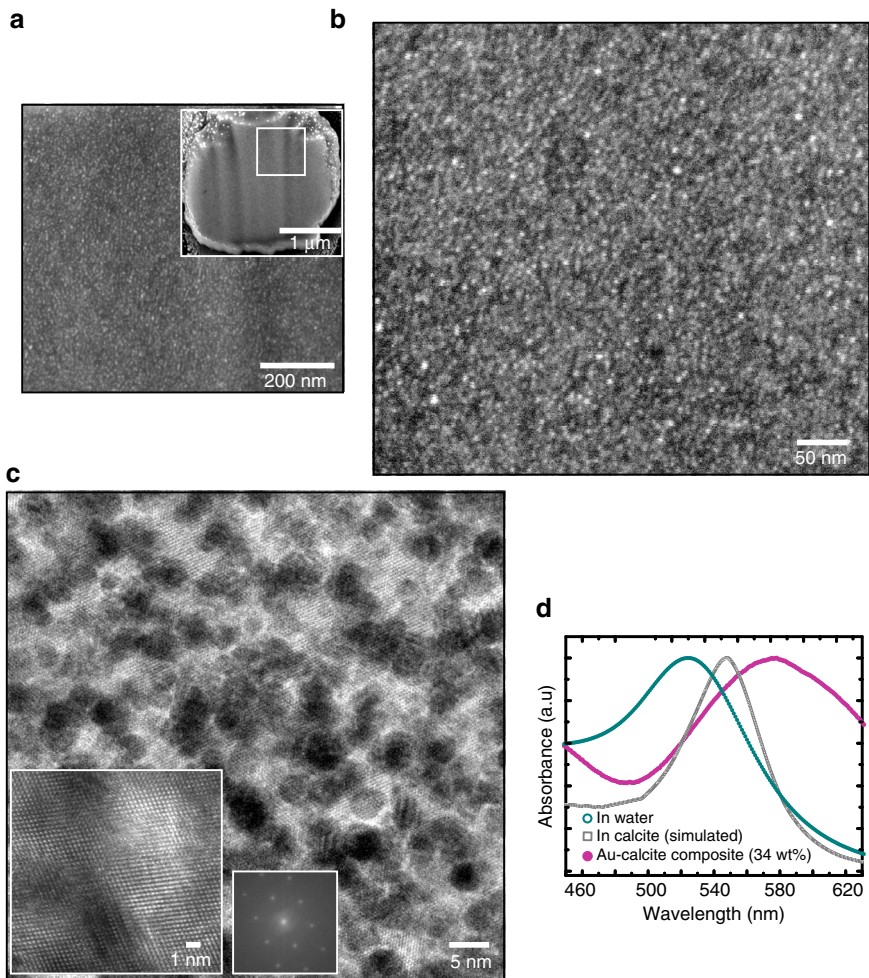

**Fig. 6** Calcite single crystals containing very high densities of 4 nm PGMA-NPs. **a**, **b** SEM images of an FIB cross-section though a crystal. **b** is a high-magnification image of **a**. **c** High-resolution bright field TEM image of a cross-section through a crystal, showing lattice fringes that demonstrate single crystallinity. The inset is an FFT of the same area. **d** UV–Vis absorption spectra comparing 4 nm PGMA-NPs in water (green), 4 nm PGMA-NPs in calcite (simulated, grey) and 4 nm PGMA-NPs in the calcite crystal.

the step edges (Supplementary Note 3). Zoning effects[31,33,34] can therefore be observed when crystals were precipitated at low supersaturations or in the presence of low concentrations of particles (see e.g. Fig. 7e). This specificity is lost as the supersaturation and/or particle concentration increases[31] such that particles are occluded homogeneously throughout the crystal.

Simulations were also performed to gain insight into the interactions between the polymer-NPs and the calcite surface. Given the structural complexity of the polymers, monomeric models of the diol and carboxylate functional motifs were employed (Fig. 10c, Supplementary Fig. 9) and their adsorption free energies at the hydrated terrace and steps were computed using well-tempered metadynamics (see Methods section for details). The monomers were found to bind with similar strengths to the terraces, and favoured the steps over the terraces, with a slight preference for the acute geometry. While the diol binds most strongly, the variation in the binding strength between the monomers is small and insufficient to account for the experimental observations. This suggests that the interactions of the functional groups with the kink sites are primarily responsible for the experimental observations. However, modelling this interaction is technically challenging and was therefore not pursued.

Additives that bind strongly to kink sites will exert a strong morphological effect[27] and should be easier to occlude due to their longer residence times on the crystal surface. As carboxylate groups are physically commensurate with carbonate ions, while the alcohols are not commensurate with calcium or carbonate ions, the former are likely to substitute more naturally into the calcite lattice and bind more strongly to kink sites. Occlusion, in turn, is strongly dependent on the growth conditions. At low supersaturations and NP concentrations, the NPs may be resident at the kink sites for long enough to induce a morphological change, but only long enough to produce weak occlusion. Under high supersaturation conditions, in contrast, the crystal can grow rapidly around an adsorbed NP with a short residence time, giving rise to effective occlusion with little morphology change. The excellent colloidal stabilities of our functionalised NPs enables us to profit from this growth regime to give high occlusion. That this can be achieved with little morphology change is consistent with the reduced preference of these NPs for the kink sites. However, even these NPs ultimately affect step propagation and change the crystal morphology at very high occlusion levels (Fig. 7g).

**Extension to alternative host crystals**. Finally, the generality of our occlusion strategy was demonstrated by extending it to calcium sulfate dihydrate (gypsum) and calcium oxalate

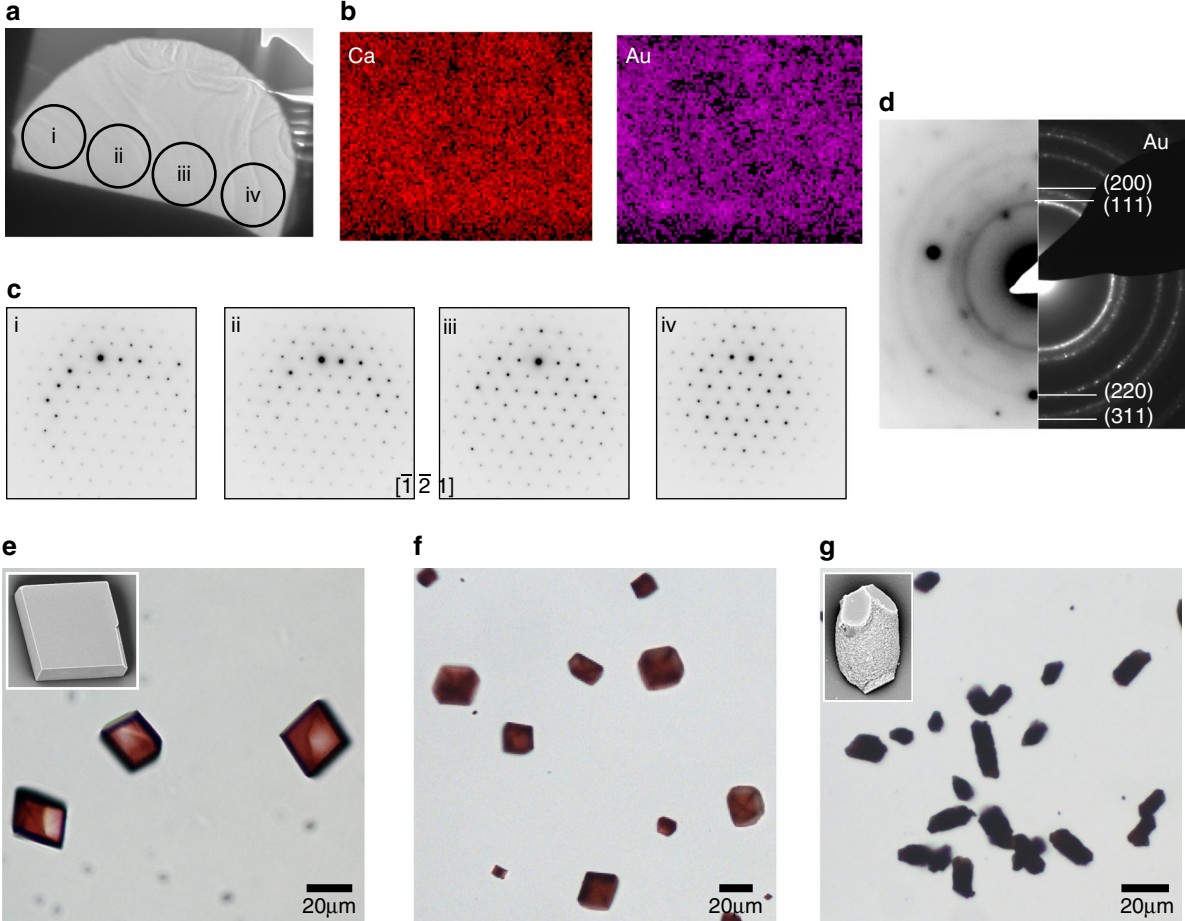

**Fig. 7** TEM and optical micrographs of calcite crystals precipitated in the presence of 4 nm PGMA-NPs. **a** Low-magnification bright field TEM image of an FIB-cross-sectioned calcite lamella showing the areas from which the selected area electron diffraction patterns (SAED) were taken. **b** STEM HAADF EDX mapping showing the homogeneous distribution of Ca and Au. **c** Corresponding SAED patterns demonstrate that single crystallinity is maintained throughout the crystal shown in **a**. **d** Off-zone axis SAED pattern showing rings corresponding to gold, which matches with diffraction pattern of Au NPs alone. **e–g** Optical micrographs of calcite crystals grown at concentrations **e** 0.001 wt%, **f** 0.005 wt%, and **g** 0.02 wt% 4 nm PGMA-NPs.

monohydrate (COM). Gypsum is considerably more soluble than calcite, and is thus precipitated from solutions with $[Ca^{2+}] > 50$ mM. It is therefore not possible to occlude NPs in gypsum at noticeable levels using acidic polymers. Calcium oxalate, in turn, is important in pathological crystallisation where its formation is believed to be regulated by highly acidic macromolecules[35]. Precipitation of COM and gypsum in the presence of PGMA-NPs generated deep red/crystals that showed strong zoning (Fig. 5 c-f). Such inter-sectorial zoning has been previously observed for dye occlusion in COM[35] and gypsum[33], and reveals the stronger binding of the PGMA-NPs to the {010} and {011} planes respectively[36].

## Discussion
Examples of the occlusion of foreign species within crystals can be found across the literature[37], and range from dyeing crystals[33] to entrapment of gels[38,39], and the formation of inclusions within rocks[40]. While weakly binding species can be occluded under an external force (as occurs in a rigid gel), occlusion of soluble additives from solution requires strong binding to the crystal surface. However, the kinetics of adsorption are also important, and strongly binding additives can induce changes in polymorph and induce polycrystallinity[37]. It is also essential that the additives maintain colloidal stability in the crystal growth solution. These

stringent criteria ensure that occlusion usually occurs at low levels, as is consistent with the widespread use of crystallisation for purification.

Calcite has been used extensively to study occlusion processes, where foreign species increasing in size from ions and small molecules[34,41,42], to NPs[43–45] and proteins[28,46,47] to sub-micron particles[48,49] have been incorporated. Systematic analysis of the occlusion of amino acids indicated that species with acidic side groups (aspartic acid and glutamic acid) occluded effectively, while moderate incorporation occurred for molecules with polar side chains[41]. Those with basic or hydrophobic chains typically exhibited very limited occlusion. However, a subsequent study showed that glycine can be occluded at levels exceeding 7 mol%, as compared with 4 mol% for aspartic acid[15], where the greater stability of glycine than aspartic acid in the crystallisation solution facilitates higher occlusion.

Incorporation of particulate species within calcite has invariably been achieved using a bio-inspired approach in which the particles are coated with polymer chains with acidic side groups[44,45,48–50]. These can be intrinsic to the particles (as for example for block copolymer vesicles)[48,50] or polymers can be adsorbed to pre-made NPs[44,51]. These systems also provide an opportunity to explore the 'design rules' underlying the occlusion of these particles, where features such as the length of the polymer chains and their packing densities on the particle surfaces can be

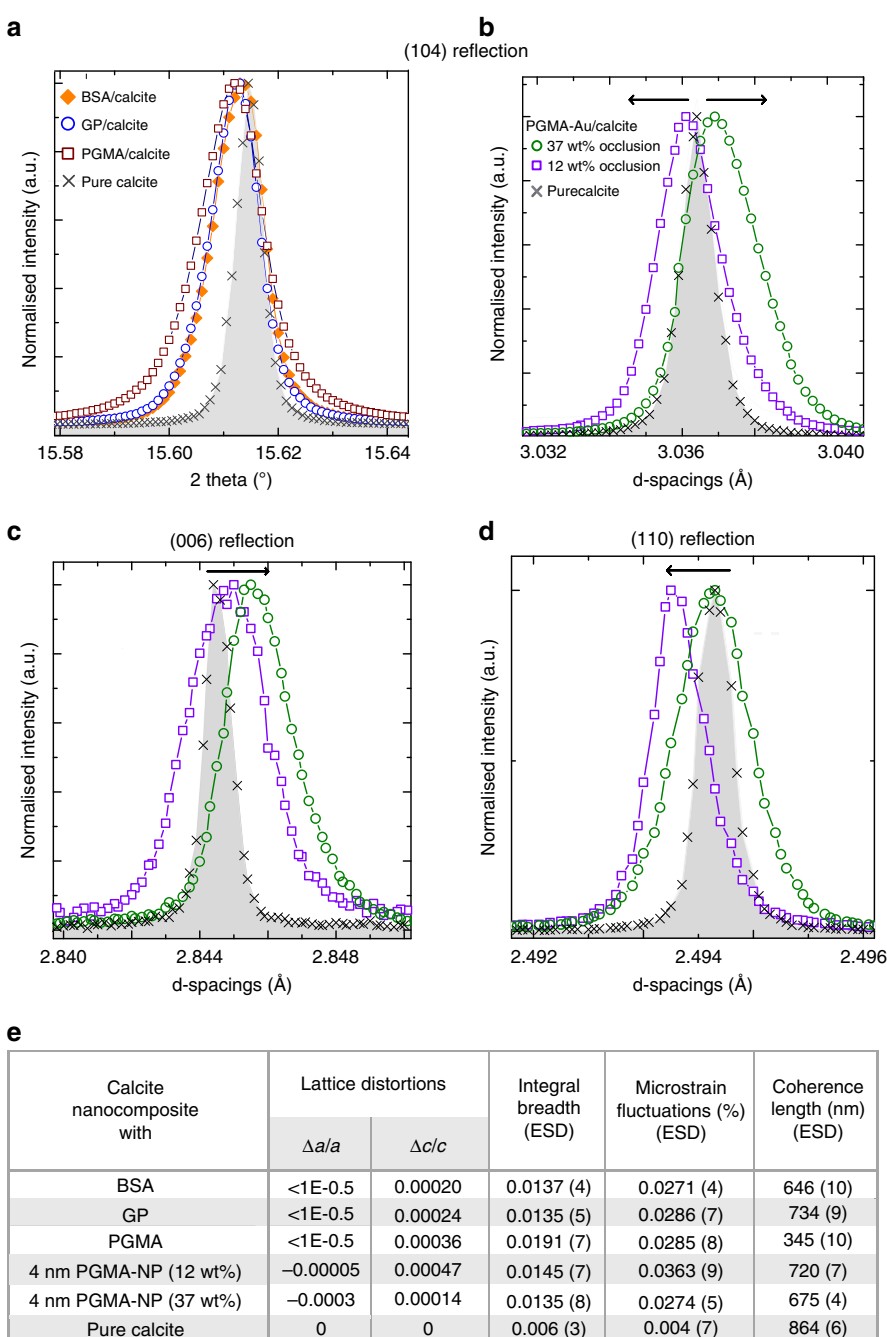

**Fig. 8** Powder XRD microstructure analysis of the protein/calcite and PGMA-NPs/calcite crystals. **a** {104} reflections of calcite crystals grown in the presence of proteins (BSA and GP) or PGMA, showing small shifts to larger d-spacings (lattice expansion) and peak broadening. **b**–**d** {104}, {006}, and {110} reflections obtained using high-resolution synchrotron powder XRD from calcite crystals with 12 wt% occlusion and 37 wt% occlusion. A shift to larger d-spacing is observed along the c-axis (expansion, seen in the {006} reflection (**c**)), and to a smaller d-spacing along the a-axis (contraction, as seen in the {110} reflection (**d**)). {104} reflections showed a transition from lattice expansion (12 wt% occlusion calcite) to contraction (37% occlusion calcite). **e** Summary of the strains and coherence lengths, where the lattice distortion and strain parameters were obtained using Rietveld analysis and line profile analysis, respectively.

varied[50,52]. The polymer chains are seen to play a dual role in which they ensure strong binding to the crystal surface, while also endowing the particles with colloidal stability in the crystallisation solution. Notably, however, these highly anionic particles invariably aggregate if present at high calcium ion concentrations, which limits the potential occlusion levels.

The synthesis of micron-sized or macroscopic nanocomposite crystals from aqueous solutions has therefore proven challenging.

In all, ≈0.01 wt% NPs have been occluded within simple salts such as NaCl or borax[53], and ≈0.15 wt% Au NPs in calcite crystals precipitated within a gel[54]. Our previous work using anionic block copolymers as NP stabilisers only incorporated 5 wt% $Fe_3O_4$[44] and 3 wt% gold NPs in calcite[51]. Examples exist of the incorporation of NPs in perovskites[55] and MOFs[56], but it is noted that these are synthesised in organic solvents that provide colloidal stability. The low-charge, hydroxyl-rich macromolecules

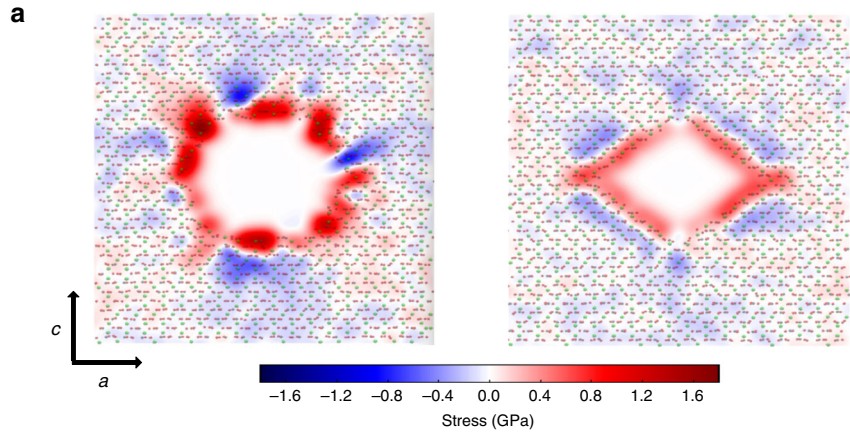

| | Pure calcite | Calcite/4 nm PGMA-Au (12 wt%) | Calcite/4 nm PGMA-Au (37 wt%) | | |
|---|---|---|---|---|---|
| | | | Experimental | Simulation Sphere | Rhombohedron |
| *a* lattice distortion (%) | 0 | −0.005 | −0.0299 | −0.0278 | −0.0334 |
| *c* lattice distortion (%) | | 0.047 | 0.0137 | −0.0145 | 0.0083 |
| Unit cell volume (Å³) | 368.03 | 368.27 | 367.97 | | |

**Fig. 9** MD simulations of the stress field associated with an occlusion in a PGMA-Au/calcite composite crystal. **a** Plots of hydrostatic stress field in calcite arising from a spherical hole (left) and rhombohedral hole (right). In each case, red represents areas under tension and blue areas under compression. The hole induces a local tension stress on the interface that is compensated for by contraction of the surrounding matrix. **b** Table comparing lattice distortions (%) and unit cell volumes obtained from experiments and simulations.

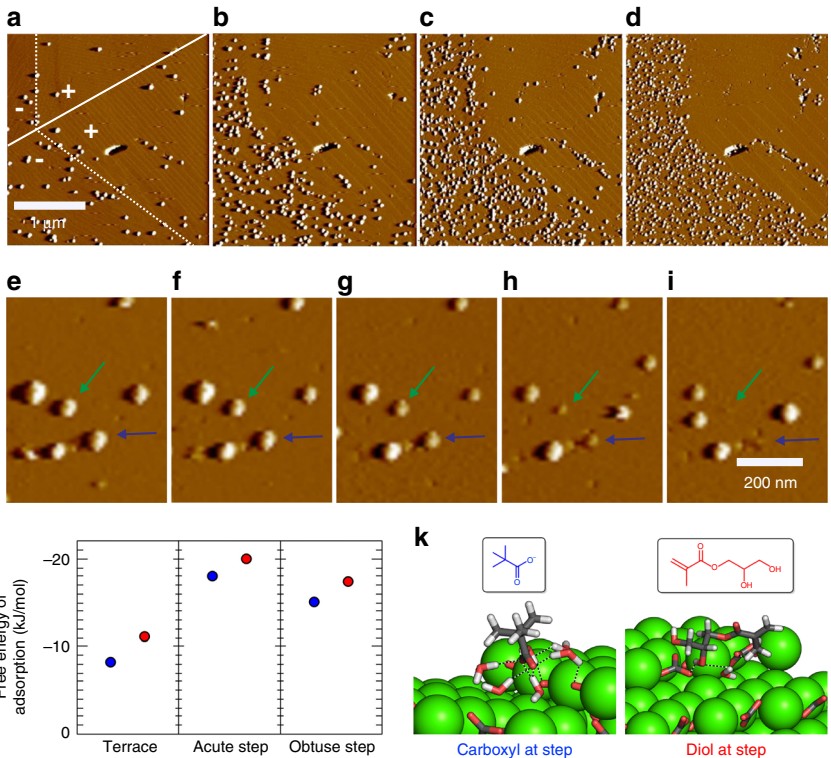

**Fig. 10** Occlusion mechanism studied by AFM and molecular dynamic simulations. AFM peak force error images of calcite crystals grown in situ in the presence of 14 nm PGMA-NPs in a liquid cell. **a–d** Time sequence images taken at **a** 4.5, **b** 9, **c** 13.5, and **d** 18 min after the introduction of the PGMA-NPs to the liquid cell. The images showed that NPs are weakly absorbed to the calcite surface with a preference for the acute step edges and surface defects. Over time there was an increase in the number density of NPs on the surface. **e–i** Time sequence images showing the incorporation of the NPs into calcite. The green and blue arrows indicate two examples of NPs that were incorporated during imaging, where images were collected in 3 min intervals. **j, k** Molecular dynamic simulations. **j** Adsorption free energies of the three molecules shown to the (104) terrace, and acute and obtuse steps. **k** The molecular configurations show the stable binding states of each molecule to the acute step. The colours are green (calcium), dark grey (carbon), red (oxygen), and light grey (hydrogen). Only the shared solvent is rendered, with hydrogen bonds shown as dashed lines.

employed here can therefore support outstanding occlusion as they bind strongly to calcite and have excellent colloidal stability under high supersaturation conditions. The latter is vital in ensuring that nanocomposites can be synthesised with a high yield.

It is interesting to note that hydroxyl-rich molecules are also widespread in $CaCO_3$ biomineralisation, where insoluble polysaccharides such as chitin fulfil structural roles. Glycoproteins are common within the soluble organic matrix of both crystalline[4] and amorphous calcium carbonate (ACC)[57], and soluble polysaccharides are present within calcium carbonate biominerals including coccoliths[58,59] and corals[60]. Notably, these molecules are also usually acidic (the coccolith polysaccharides strongly so), making it difficult to distinguish between the activities of the acidic and hydroxyl groups. However, the acidic groups appear to dominate in the ability of these molecules to inhibit crystal growth[61], while the uncharged saccharide groups modulate their interaction with the crystal surface[4,37]. This is also consistent with studies showing that small alcohol molecules bind so strongly to calcite that they displace water molecules[23], and that stable ACC can be formed in absolute ethanol[62].

Finally, this synergistic relationship between the acidic and hydroxyl-rich side groups is also seen in experiments and modelling with oligopeptides[63,64] and synthetic peptoids[65], and in particular by screening peptidic phage display libraries[19–21,66]. These have produced lists of peptides, many of which were not particularly acidic. Deca-peptides that bound strongly to calcite and aragonite only contained a single aspartate or glutamate residue, together with basic and neutral amino acids[19], while calcite-binding 15-mers were enriched in aspartic/glutamic acid and serine/threonine[20]. Seven-mers that bound strongly to vaterite were again rich in residues with hydroxyls[21].

In summary, the influence of organic additives on calcium carbonate precipitation is traditionally evaluated in terms of changes in morphology, polymorph and orientation, which has left low-charge molecules under-explored. Our results demonstrate that low-charge, hydroxyl-rich macromolecules can facilitate the synthesis of a unique class of materials—single crystal nanocomposites—giving exceptional control over their compositions and structures. The use of generic proteins and simple homopolymers makes the approach readily accessible and scalable. With the ability to control NP loadings over such a wide range, it is also possible to engineer the strain within the host crystal from a lattice expansion to contraction. Such lattice strain is intimately linked to a range of electronic, optical, magnetic, mechanical and thermal properties[67–69]. This methodology therefore has enormous potential for the synthesis of nanocomposites for applications such as thermoelectrics, optoelectronics and catalysis[67,70], and with the enormous world-wide use of calcite carbonate in, for example, paints, coatings and drug delivery systems it is expected to find immediate utility.

## Methods

**Materials for polymer synthesis.** Methacrylic acid (MAA), 4,4′-azobis(4-cyanopentanoic acid) (ACVA), 2-cyano-2-propyldithiobenzoate (CPDB) and 4-cyano-4-(phenylcarbonothioylthio)pentanoic acid (CPCP) were purchased from Sigma-Aldrich (UK) and were used without further purification. Glycerol monomethacrylate (GMA; 99.8 %) was donated by GEO Specialty Chemicals (Hythe, UK) and was used without further purification.

**Materials for commercial proteins and polymers.** Bovine serum albumin (BSA, Mw 66.4 kDa, single polypeptide chain with 583 amino acid residues and no carbohydrates), alpha-1-acid glycoprotein (GP, from bovine plasma, Mw 41 kDa, total carbohydrate 45%, 11–12% sialic acid), poly(2-hydroxyethyl methacrylate) (poly(HEMA) Mw 20,000), poly(vinyl alcohol) (PVA, Mw 9000–10,000, 80% hydrolysed), poly(allylamine hydrochloride) (PAH, Mw 15,000), ethylenediamine and 1-ethyl-₃-(3-dimethylamino-propyl)-carbodiimide (EDC) were purchased from Sigma-Aldrich (UK) and were used without further purification.

**Materials for inorganic synthesis.** Gold(III) chloride trihydrate, sodium borohydride, sodium citrate, $CaCl_2 \cdot 2H_2O$, $(NH_4)_2CO_3$, $CaCO_3$, $Na_2SO_4$ and $Na_2C_2O_4$ were purchased from Sigma-Aldrich (UK) and were used without further purification.

**Synthesis of PMAA.** A round-bottomed flask was charged with methacrylic acid (MAA) (5.00 g; 58.1 mmol), 4-cyano-4-(phenylcarbonothioylthio)pentanoic acid (CPCP) RAFT agent (232.0 mg, 0.83 mmol), ACVA (47.0 mg; 0.166 mmol; [CPCP]/[ACVA] = 5) and ethanol (6.35 g). The flask was sealed and purged with nitrogen gas for 30 min and was placed in a pre-heated oil bath at 70 °C. The reaction was then allowed for 3 h and was quenched by immersing the flask in an ice bath and exposing the reaction mixture to air, and the crude PMAA macro-RAFT agent was purified by dialysis, first against 1:1 deionised water/methanol and then against DI water. The PMAA was subsequently obtained by freeze-drying. A full conversion (>99 %) of the monomer and a mean DP of 70 ($PMAA_{70}$) was estimated using $^1H$ NMR spectroscopy performed in $D_2O$ by comparison of the integrated signal intensity of the aromatic protons of the RAFT agent at 7.4–7.9 ppm with that of the methacrylic polymer backbone protons at 0.5–2.5 ppm. PMAA was exhaustively methylated using excess trimethylsilyldiazomethane prior to THF GPC analysis. Values of Mn = 10,000 g $mol^{-1}$ and Mw/Mn = 1.18 were obtained.

**Synthesis of PGMA.** Glycerol monomethacrylate (GMA) (5.00 g, 31.2 mmol), 2-cyano-2-propyldithiobenzoate (RAFT agent, CPDB, 137.9 mg, 0.62 mmol), ACVA (35.0 mg; 0.12 mmol; [CPDB]/[ACVA] = 5) and ethanol (6.00 g) were added to a round-bottomed flask to target a mean degree of polymerisation (DP) of 50. The flask was sealed and purged with nitrogen gas for 30 min, and was placed in a pre-heated oil bath at 70 °C. The reaction was allowed to proceed for 3 h, and was then quenched by immersing the flask in an ice bath and exposing the reaction mixture to air. The crude PGMA macro-RAFT agent was purified by precipitation using a 10-fold excess of dichloromethane (DCM) and was subsequently dissolved in water and then isolated by freeze-drying. A mean degree of polymerisation (DP) of 58 ($PGMA_{58}$) was estimated using $^1H$ NMR spectroscopy performed in $D_2O$ by comparison of the integrated signal intensity of the aromatic protons of the RAFT agent at 7.4–7.9 ppm with that of the methacrylic polymer backbone protons at 0.5–2.5 ppm. Considering the target and obtained DPs of 50 and 58, respectively, and a GMA monomer conversion of 85%, this suggests a chain transfer agent efficiency of 73%. GPC performed in DMF indicated Mn = 14,000 g $mol^{-1}$ and Mw/Mn = 1.14.

**$^1H$ NMR spectroscopy.** $^1H$ NMR spectra in $D_2O$ were recorded using a Bruker Avance 400 spectrometer operating at 400 MHz using $CD_3OD$ as the solvent (Supplementary Fig. 10).

**APC and GPC analysis.** The molecular weight distributions of PGMA were determined using a Waters ACQUITY APC (advanced polymer chromatography) system equipped with an ACQUITY refractive index (ACQRI) detector and ACQUITY APC XT (200 Å, 2.5 μm) column. The column temperature was maintained at 40 °C and the APC eluent was DMF with 10 mM LiCl, at a flow rate 0.5 mL $min^{-1}$. The system was calibrated using a series of ten, near-mondisperse poly(methyl methacrylate) standards (Mn = $6.25 \times 10^2$–$6.18 \times 10^5$ g $mol^{-1}$, $K = 2.094 \times 10^{-3}$, $\alpha = 0.642$).

The molecular weight distributions of PMAA were determined by THF GPC (gel permeation chromatography) analysis, which was carried out using an Agilent Technologies Infinity 1260 set-up equipped with a refractive index detector and two PL aquagel-OH 30 8 μm columns running at 35 °C. The GPC eluent was THF containing 2 v/v % triethylamine and 0.05 wt/v% butylhydroxytoluene (BHT) at a flow rate of 1.0 mL $min^{-1}$. Calibration was conducted using a series of ten near-monodisperse poly(methyl methacrylate) standards (Mn = $6.25 \times 10^2$–$6.18 \times 10^5$ g $mol^{-1}$, $K = 2.094 \times 10^{-3}$, $\alpha = 0.642$).

**Modification of BSA via amination.** Anionic residues including aspartic acid and glutamic acid contribute to the negative charge of the BSA protein. These were converted into cationic groups by reacting the BSA with ethylenediamine in the presence of 1-ethyl-3-(3-dimethylamino-propyl)-carbodiimide (EDC) as a catalyst following a published protocol[71]. Five hundred milligrams BSA and 50 mg EDC were fully dissolved in 100 mL PBS buffer solution (pH 7.4) and were then transferred to a 250 mL round-bottomed flask, where 50 ml of ethylenediamine (1 M) was added during magnetic stirring. The reaction was allowed to proceed for 2 h with stirring at room temperature, and was then purified by dialysis against DI water for 2 days, followed by freeze-drying. This treatment resulted in a change in the zeta potential from −16 to −5 mV at pH 9.

**Removal of sialic acid from alpha-1-acid glycoprotein.** The high sialic acid content of alpha-1-acid glycoprotein (11–12%) is the principal component of its acidic nature. The sialic acid groups were therefore removed using a standard hydrolysis treatment[72]. One hundred milligrams α-1-acid glycoprotein was dissolved in 50 mL $H_2SO_4$ (12.5 mM) solution in a 100 mL round-bottomed flask, and

the solution was then heated at 80 °C for 1 h with stirring. The solution was cooled to room temperature and was then neutralised with $Ba(OH)_2$ to give a pH of 6. The solution was then dialysed against water for 2 days, followed by freeze-drying. This treatment resulted in a change in the zeta potential of the NPs from −6 to 6 mV at pH 9.

**Synthesis of polymer/protein-coated gold nanoparticles**. Au nanoparticles were synthesised using the Turkevich-Frens method[73]. Four nanometers Au NPs were synthesised by adding 4 mL of 1 wt% $HAuCl_4$ $3H_2O$ (25.4 mM) to 360 mL DI water, and then adding 8 mL of a 1 wt% solution of sodium citrate (38.2 mM). The solution was stirred for 5 min, before quickly adding 4 mL of 20 mM sodium borohydride solution with vigorous stirring. The solution immediately turned brownish-red and was stirred for a further 30 min. Fourteen nanometers Au NPs were synthesised by placing 400 mL of DI water in a 500 mL round-bottomed flask fitted with a condenser, and heating the solution to reflux. Twenty milliliters of a 1 wt% solution of sodium citrate (38.2 mM) were then added under vigorous stirring, and the colour changed to transparent to purple and then ruby red. The solution was stirred for a further 20 min and was then cooled to room temperature.

The gold NPs were coated with proteins or polymers by adding 40 mg of the desired polymers or proteins to the prepared solutions of 14 nm Au NPs, and 80 mg of the desired polymers to the prepared solutions of 4 nm Au NPs, and stirring overnight at room temperature. The solution was then subjected to multiple concentration/dilution cycles using an Amicon® stirred ultrafiltration cell (Millipore) with 100 or 300 kDa membranes, maintaining the pressure below 30 psi (2 bar). The prepared gold NP solutions were further adjusted to 0.1 wt% by measuring the absorbance at 450 nm using UV spectroscopy. Extinction coefficients (molar absorptivity) of $3.6 \times 10^6$ $M^{-1}$ $cm^{-1}$ for 4 nm and $1.76 \times 10^8$ $M^{-1}$ $cm^{-1}$ for 14 nm gold NPs were employed[74].

**Synthesis of PAH-stabilised gold NPs**. Amine stabilised Au NPs could not be made by the typical Turkevich-Frens method as addition of positively charged poly (allylamine hydrochloride) (PAH) caused immediate aggregation of the citrate-stabilised Au NPs. PAH-stabilised Au NPs were therefore synthesised using an alternative published protocol[75]. The particle size is limited to 5 nm in this method. Four milliliters of $HAuCl_4$ $3H_2O$ (25.4 mM) were dissolved in 380 mL of DI water in a 500 mL round-bottomed flask fitted with a condenser, and the solution was heated to reflux with stirring. One hundred milligrams of PAH were then added, the pH of the solution was adjusted to 9 using 100 mM $Na_2CO_3$ solution and it was stirred overnight at room temperature. The solution was then concentrated using an ultrafiltration cell as described above.

**Characterisation of polymer/protein-stabilised NPs**. Diluted solutions of the protein- or polymer-stabilised NPs were investigated using a Perkin Elmer Lambda 35 UV–vis double-beam spectrometer, where an extinction peak at ≈525–526 nm corresponds to the gold surface plasmon. The sizes of dried NPs were determined using TEM, where samples were prepared by placing a droplet of a suspension of the NPs on a carbon-coated TEM grid. They were then imaged using a FEI Tecnai TF20 FEG-TEM fitted with an Oxford Instruments EDX system and operating at 200 kV. Thermal gravimetric analysis (TGA) using a TA Instruments Q600 SDT was carried out to estimate the ratio of the polymer/protein and Au NPs. The analysis was performed at a 10 °C $min^{-1}$ heating rate from room temperature to 800 °C in air.

The zeta potentials and hydrodynamic diameters of the NPs were measured using dynamic light scattering (DLS) and aqueous electrophoresis. This was carried out using a Malvern Zetasizer NanoZS instrument equipped with a 4 mW He–Ne laser operating at 633 nm, an avalanche photodiode detector with high quantum efficiency and an ALV/LSE-5003 multiple tau digital correlator electronics system. Aqueous solutions of concentrations 0.02 wt% Au NPs in 20 mM $CaCl_2$ that had been adjusted to pH 9 using 100 mM NaOH were analysed. DLS was performed by detecting back-scattered light at an angle of 173° using disposable PMMA cells. Electrophoresis measurements were conducted using a disposable capillary cell (Malvern).

**Ammonium carbonate diffusion method**. Calcium carbonate was precipitated using the ammonium carbonate diffusion method[24] in the presence of soluble additives (protein/polymer-stabilised NPs, polymers or proteins) This was typically carried out using 4 or 24 multi-well plates. Each well contained a total of 1 mL of the crystallisation solution, and pre-cleaned glass slides were placed at the base of the each well as substrates for crystal growth. Stock solutions of $[Ca^{2+}] = 10$ or 100 mM and 0.1 wt% solutions of Au NPs stabilised with proteins or polymers were mixed with DI water to obtain final concentrations of $[Ca^{2+}] = 1.25–50$ mM and NP concentrations of 0–0.05 wt%. The plate was then placed in a sealed dessicator with a Petri dish containing 2.0 g of solid $(NH_4)_2CO_3$ that was covered with Parafilm that had been punctured with a needle. Crystallisation was typically allowed to proceed overnight (16 h). Following this period, the glass slides supporting the $CaCO_3$ crystals were removed from solution, washed with DI water and then ethanol, and dried at room temperature. Larger quantities of crystals were prepared for PXRD and AAS by scaling up the reaction to a 20 mL volume in large petri dishes.

**Kitano method**. Five hundred milligrams of $CaCO_3$ powder was added to 500 mL DI water, and $CO_2$ was bubbled through the solution for 2 h under stirring until the pH of the solution reached a value of 6 (ref. [76]). The solution was then filtered through a 200 nm syringe filter (Millipore) to remove undissolved calcium carbonate powder. This solution was then mixed with the solution of Au NPs, and crystallisation was allowed to proceed overnight in air.

**Precipitation of calcium sulfate and calcium oxalate**. Stock aqueous solutions of 1 M $CaCl_2$, 1 M $Na_2SO_4$, and 0.3 wt% gold NPs were prepared. Gypsum ($CaSO_4$ $2H_2O$) was precipitated by mixing the calcium and sulfate solutions to give a solution of composition 100 mM $CaCl_2$, 0.08 wt% PGMA stabilised Au NPs and 200 mM $Na_2SO_4$ solution in 4 or 24 multi-well plates. Calcium oxalate monohydrate (COM) was precipitated by mixing equal volumes of a 1.5 mM $CaCl_2$ solution containing 0.02 wt% gold NPs and a 1.5 mM sodium oxalate solution in 4 or 24 multi-well plates.

**General characterisation of crystals**. The morphologies and colour of the crystals precipitated on glass slides were analysed using optical microscopy with a Nikon Eclipse LV100 polarising microscope equipped with transmitted and reflected light sources. Routine identification of crystal polymorph was carried out using Raman microscopy and PXRD. Raman spectra were recorded using a Renishaw Raman Microscope equipped with a 785 nm laser, while laboratory PXRD experiments were carried out using a PANalytical X'Pert[3] diffractometer equipped with a Cu-anode ($\lambda = 1.54056$ Å).

**SEM and focused ion beam milling**. SEM was used to image the crystals, where samples were prepared by mounting the glass slides on Al SEM stubs with silver paste and coating with 10 nm Ir. The distribution of NPs within these calcite crystals was determined using focused ion beam (FIB) milling to generate a cross-section. FIB was performed using an FEI Helios G4 CX dual beam-high-resolution monochromated FEG-SEM with a precise FIB system, equipped with Ga-beam and a field emission electron gun. An area (1 μm in width) in the middle of a crystal was pre-coated with Pt (2 μm in thickness). Ga-ion currents from 2.3–7 nA, 30 kV were applied to mill up to the middle of the crystal. A final polishing/cleaning procedure was carried out at 0.8 nA and 30 kV. The resulting crystal cross-section was then imaged with a CBS concentric (insertable) higher energy electron detector without further coating of the sample.

**Transmission electron microscopy**. The structure of the calcite/NP composite crystals and the distribution of the Au NPs was investigated using TEM and associated SAED using an FEI Tecnai TF20 FEG-TEM field emission gun TEM/STEM operating at 200 kV and fitted with an HAADF detector, an Oxford Instruments INCA 350 EDX system/80 mm X-Max SDD detector and a Gatan Orius SC600A CCD camera. Thin lamellae were prepared from the nanocomposite crystals as described above, and lift-out was performed in situ using a Keindiek micromanipulator, to transfer the lamella to a copper TEM grid. The distribution of Au NPs was further confirmed using a HAADF-STEM imaging mode, and energy dispersive X-ray (EDX) analysis mapping of Ca and Au was performed. The collection angle of the HAADF-STEM imaging was 80–240 mrad. Gold NPs were also characterised using TEM, where samples were prepared by placing a couple of drops of the NP solution on carbon-coated TEM grids, and allowing them to dry in air.

**Cryo-electron tomography**. Cryo-electron tomography was performed using a Thermo Fisher Scientific Titan Krios microscope operating at 300 kV using a BioQuantum-K2 summit direct electron detector. The energy filter slit width was 20 eV and the tilt series were collected using TEM Tomography (Thermo Fisher Scientific). Tilt series were collected at a −0.5 μm defocus between ±60°, starting at 0° with a 2° increment. Lamellae were imaged at nominal magnifications of 33,000, resulting in a calibrated pixel size of 4.28 Å. Tilt images were recorded in counting mode with a dose per image of $1.52e$ $Å^{-1}$, and a total dose of $92.7e^-$ $Å^{-1}$.

**STEM tomography**. This was carried out using an EL Titan3 Themis 300 X-FEG 300 kV S/TEM with S-TWIN objective lens, monochromator (energy spread approx. 0.25 eV), multiple HAADF/ADF/BF STEM detectors, FEI Super-X 4-detector EDX system and a Gatan OneView 4 K CMOS digital camera. Tilt series were collected between ±60°, starting at 0° with a 1° increment.

**Image processing and segmentation**. Tilt series were reconstructed into tomograms using the Etomo suite in IMOD. Images with unacceptable charging, movement or poor focus were discarded. Tilt series were aligned using the gold particles in the lamella. Tomograms were reconstructed using images binned by 4. Images were segmented using Amira (Thermo Fisher Scientific) and a median filter was applied to aid thresholding of the data. The 'Interactive Thresholding Tool' was used to segment the data, and the 'remove small spots' functionality was used to remove noise from the segmentation. Supplementary Videos 1 and 3 show the raw tomograms, followed by the filtered data, then 2D segmentation of the data. Finally, the data are shown rendered in 3D.

**Atomic absorption spectroscopy**. The quantities of Au NPs in the calcite crystals were determined using AAS with a Perkin Elmer Atomic Absorption Spectrometer AAnalyst 400, operating with an air-acetylene flame. All calcite nanocomposite samples were measured after being bleached in 4 w/v % sodium hypochlorite for 48 h to remove surface-bound particles. After the bleaching process, the samples were thoroughly washed with water, then ethanol, before drying in air. Samples for AAS were prepared by dissolving the nanocomposite crystals in 200 μL of aqua regia solution made by mixing HCl and $HNO_3$ in a 3:1 ratio, then diluting the solution to 20 mL with DI water. The amounts of Au and Ca present within the same solution were then measured after calibration with 1, 2, 5 and 10 μg mL$^{-1}$ Ca and Au standard solutions.

**PXRD analysis**. PXRD measurements were carried out using a PANalytical X'Pert[3] diffractometer equipped with a Cu-anode ($\lambda = 1.54056$ Å). Sample powders were gently ground, and then loaded onto a Si wafer sample holder. The samples were rotated at a rate of 60 r.p.s. and were measured using a 0.02 step and 0.5 s exposure time. Quantification of Au in nanocomposite crystals was estimated using Rietveld analysis using PANalytical X'Pert HighScore Plus software.

**Synchrotron high-resolution PXRD analysis**. The high-resolution PXRD measurements were carried out on beamline I11 at Diamond Light Source Ltd, Didcot, UK. Instrument calibration and wavelength refinement ($\lambda = 0.824681(10)$ and $0.824677(10)$ Å) were performed using high-quality NIST silicon powder (SRM640c) and instrumental contribution to the peak widths does not exceed 0.004°(ref.[77]). Diffractograms were recorded from the specimens at room temperature. Sample powders for analysis were loaded into 0.5 mm borosilicate glass capillaries, and to avoid intensity spikes from individual crystallites, the samples were rotated during measurements at a rate of 60 r.p.s. PXRD data were then obtained using high-resolution MAC (multi-analyser crystal) diffraction scans with scan times of 1800 s.

The structural parameters were refined by Rietveld analysis using PANalytical X'Pert HighScore Plus software. Lattice parameters were determined using Rietveld analysis for whole spectrum and strain and size analysis were performed using line profile analysis for {104} {110} and {006} reflections. Peak broadening was expressed as integral breadth, which is the width of a rectangle that can be placed within the peak that has the same area as the net peak area (net area/net height). Microstrain (%) and coherence length (nm) are derived using line profile analysis.

**Single crystal diffractometry**. Measurements were carried out at 120 K using a Rigaku SuperNova diffractometer equipped with an Atlas CCD detector and connected to an Oxford Cryostream low-temperature device using mono-chromated Mo $K_\alpha$ radiation ($\lambda = 0.7107$ Å) from a Microfocus X-ray source. Diffraction images were processed and visualised using CrysAlisPro software.

**Liquid cell AFM**. In situ AFM analysis was performed using established procedures[45]. Freshly made calcite seed crystals ($\approx 50$ μm) were precipitated on glass coverslips that had been cleaned with piranha solution such that they expressed fresh {104} surfaces. These coverslips were glued to the AFM specimen disk. A commercial fluid cell (MTFML; Veeco Probes) with O-ring was placed on the coverslip inside an AFM (Bruker Multimode 8 with NanoScope V controller). A syringe pump was used to flow growth solutions through the fluid cell, and imaging was performed at room temperature using commercially available cantilevers. Bruker SNL-10 tips with nominal spring constants of 0.35 N m$^{-1}$ were used for contact mode or tapping mode imaging, and Bruker ScanAsyst-Fluid tips with nominal spring constants of 0.7 N m$^{-1}$ were used for imaging in Peak Force mode. The in situ images of the calcite surface were obtained principally in Peak Force Tapping mode at solution flow rates of 0.3 mL min$^{-1}$.

The growth solutions were prepared by combining aliquots from pre-prepared stock solutions of known concentrations. The carbonate source in these experiments was NaHCO$_3$, and these stock solutions were freshly prepared prior to experiments. The calcium source was CaCl$_2$, and the ionic strength was adjusted using NaCl. The final concentrations were [Ca$^{2+}$] = 1–2 mM, [HCO$_3^-$] = 2–4 mM and [NaCl] = 30–50 mM. The concentration of PGMA-NPs and GP-NPs used in these experiments was 0.00075 and 0.0005 wt%, respectively.

**Free energy calculations**. For each simulation, the configuration was composed of a calcite slab plus a water layer that contained the organic molecule of interest. The simulation cell was periodic in all three dimensions. For the simulations of the organic molecule interacting with a terrace, the calcite slab consisted of six layers and was approximately 40 Å in width. For the simulations with steps, the slab consisted of five complete layers and a sixth partial layer that exposed the acute and obtuse steps. The partial layer was made of six ionic rows, and the steps were separated by 31 Å. In all cases, the two surfaces were separated by approximately 50 Å, with water in-between. The simulations were performed with the LAMMPS molecular dynamics code[78]. After equilibrating the pressure to 1 atm, the system was integrated in the NVT ensemble with an integration time step of 1 fs. A

temperature of 300 K was maintained by a Nosé–Hoover thermostat with a time constant of 0.1 ps.

The CaCO$_3$ and CaCO$_3$–water interactions were taken from Raiteri et al.[79]. The water was modelled with the flexible SPC/FW model[80]. All interactions involving the organics came from the Generalised Amber Force Field (GAFF)[81] and the cross-terms were as described in Nalbach et al.[82]. It should be noted that, unlike the CaCO$_3$ interactions, the organic force fields have not been validated against experimental or ab initio thermodynamic data for these organic molecules. However, there is precedent for using this particular combination of force fields to model calcite/water/organic systems[82].

The free energy maps were computed using metadynamics, as implemented in PLUMED[83]. For organic-terrace interactions, the distance between the calcite surface and the organic molecule, normal to the surface ($z$), was used as the collective variable. While for the organic-step interactions, the distances normal to the surface ($z$) and orthogonal to the step ($y$) were taken as the collective variables. For the carboxylated molecule, the position of the carbon atom in the carboxyl group was taken as the reference point, while for the larger alcohol molecules, the centre of mass was taken as the reference. The organic molecules were confined with soft walls to within 15 Å of the surface along $z$ and, for the steps, to within 10 Å of each step along $y$. The metadynamics simulations deposited Gaussians of width 0.2 Å and an initial height of kT. The process was accelerated using multiple walkers[84] and the well-tempered scheme[85], with a bias factor of 7. The terrace (step) simulations were performed for an aggregated time of 100 ns (300 ns).

The cations in calcite are known to hydrate strongly, giving rise to water molecules at steps with long residence times in excess of $\approx 10$ ns[86]. Consequently, if the binding of a molecule to the calcite step involves the dehydration of the cations, then the dehydration process must also be driven by the collective variables to faithfully reconstruct the free energy surface[87]. However, in our simulations, we found that none of the binding states involved direct binding to the cations. The only direct binding was between the alcohol groups and the carbonates. It was therefore unnecessary to drive dehydration of the steps. This point was further substantiated by initiating the configurations with the molecules adsorbed directly to a dehydrated step prior to introducing water; upon relaxation, the adsorbates transitioned to more of an outer-sphere adsorption state within the 10 ps timescale. The resulting free energy maps for the terraces and steps are shown in Supplementary Fig. 9.

**Structural calculations**. For the stress and lattice parameter calculations, a hexagonal simulation box was used with dimensions equal to 18, 18 and 6 repetitions of the $a$, $b$ and $c$ components of the calcite unit cell respectively. This resulted in box sizes equivalent to roughly 10 nm along each direction and periodic boundaries were used in every dimension. A hole, with either a spherical geometry or a rhombohedral geometry, was cut from the centre of the simulation box. The spherical hole was given a diameter of 4 nm, and charge neutrality was ensured via the random removal of excess ions at the surface of the hole. The size of the rhombohedral hole was calibrated such that the volume of the rhombohedron was similar to that of the sphere.

Both geometries were simulated using LAMMPS[78], and compared with a control simulation with no hole present. The force fields used for calcite were those obtained by Raiteri et al.[79]. A time step of 1 fs was used to integrate the equations of motion. Nosé-Hoover thermostats and barostats, with timescales of 0.1 and 1 ps, respectively, were employed to control the temperature at 300 K and the pressure at 1 atm. All simulations consisted of an equilibration period of 500 ps, followed by a 2.5 ns period of collecting and averaging data. The hydrostatic stress distributions were calculated using the atomic virials, time-averaged over the final 2.5 ns simulation. From these virials, stress distributions were estimated using the method implemented by Darkins et al.[88]. The average lattice spacings in the $a$ and $c$ directions were determined using projections of the radial distribution function onto the $a$ and $c$ planes. A perpendicular cut-off of 1.5 Å for this RDF was used to account for any thermal noise and crystal distortions, while preventing any adjacent atoms from contributing to the RDF. The final results were probability distributions for the $a$ and $c$ spacings, which were averaged to obtain the lattice parameters.

## Data availability

Additional data that support the findings of this study are available in the Research Data Leeds Repository with the identifier [https://doi.org/10.5518/733]

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

## Acknowledgements

This work was supported by an Engineering and Physical Sciences Research Council (EPSRC) Platform Grant (to F.C.M., M.A.H. and Y.-Y.K., EP/H005374/1), an EPSRC Programme Grant (EP/R018820/1, to F.C.M., Y.-Y.K., R.D., A.B., and D.M.D.), an EPSRC research grant (EP/P005233/1) and the EPSRC 'Complex Particulates' CDT (to O.N.). We acknowledge Diamond Light Source for time on beamline I11, and Dr. Nicole Hondow and Dr. Stuart Micklethwaite in Leeds Electron Microscopy and Spectroscopy Centre (LEMAS) for help with STEM tomography and FIB imaging, and we thank Dr. Paul Thornton, Chemistry, Leeds, for advice regarding the polymer synthesis and APC analysis. We are grateful to YiFei Xu and Christopher Pask, Chemistry, Leeds, for helping with the tomography imaging analysis and single crystal diffraction analysis, respectively. We thank GEO Specialty Chemicals (Hythe, UK) for kindly donating the glycerol monomethacrylate. The Titan Krios microscopes used in this work were funded by the University of Leeds (UoL ABSL award) and the Wellcome Trust (108466/Z/15/Z).

## Author contributions

F.C.M. and Y.-Y.K. originated the project. Y.-Y.K. led the experimental work, preparing all samples and carrying out microscopy and spectroscopy analysis. R.D., A.B. and D.M.D. carried out the simulations, while C.C.T. and Y.-Y.K. conducted the PXRD experiments and Y.-Y.K. analysed the data. R.F.T. carried out the cryo-electron tomography and associated image processing, while A.N.K. performed the FIB and SEM work. M.A.H. performed the AFM work and analysed the data. O.N. synthesised and carried out the polymer characterisations. F.M. contributed to the discussion of biomineralisation proteins, and S.P.A. commented on the manuscript. All authors contributed to the preparation of the manuscript.

## Competing interests

The authors declare no competing interests.
