## [Peer Review File · Nature Communications]

REVIEWERS' COMMENTS:

Reviewer #1 (Remarks to the Author):

This interesting paper has much improved on revision. I support publication in Nature Communication and have only a few minute details to be addressed

1. On line 129, the authors give D_a and D_c for the lattice expansion, please also give the associated standard uncertainty (standard deviation)
2. On page 10 line 325, there are 4 periods at the end of the line
3. In Figure 8e, the authors give microstructure analysis and list microstrain fluctuations under a heading "macrostrain fluctuation"; please correct to "microstrain fluctuations"

Reviewer #2 (Remarks to the Author):

The reported results are interesting, especially for the surprisingly large amount of nanoparticles occluded in the crystals. However, the possible mechanism and the potential application for this surprisingly phenomenon are not clearly described in this paper.

What is the possible driving force to highly concentrate the nanoparticles into the crystals? The MD simulations are based on the monomeric models and they do not show the reason for the nanoparticles or even the organics to be concentrated into the crystals. Assuming strong affinity between the organics and the crystals is the reason, it is naturally to expect that the organics (without the nanoparticles) should be concentrated into the crystals that grow from the solution of the organics. But there is no evidence for it.

Reviewer #3 (Remarks to the Author):

With regard to the occlusion of nanoparticles into calcite, I find the authors' proposed mechanism plausible and well-described, and consistent with prevailing ideas on how calcite grows from solution. The authors support their proposed mechanism with evidence from experiments and computer simulations. While mechanisms that are different from the proposed one cannot be ruled out completely based on the evidence presented, a substantial amount of additional computational work would be needed to nail down all the details. Such work clearly exceeds the scope of this paper. It is unclear to me what additional experiments could be performed to corroborate the proposed mechanism. (I note that reviewer 3, who is concerned about the mechanism, does not suggest any additional work that could clarify the point.)

Overall, it is my opinion that the manuscript in its current form presents a highly interesting study that will get the attention of a broad community of researchers. That nanoparticles can be occluded in calcite at such high concentrations and in such a straightforward manner is highly surprising to me. Since I first saw the manuscript, the authors have improved it markedly through several rounds of review, adding substantial new results and characterization in response to reviewers' requests. While a crystal-clear mechanistic picture would be of course desirable, it is not an essential question that should determine the fate of the manuscript. My own conclusion is that this is a high-quality study that clearly deserves publication in Nat. Communications.

Response to Reviewers

Reviewer #1: This interesting paper has much improved on revision. I support publication in Nature Communication and have only a few minute details to be addressed.

1. *On line 129, the authors give D_a and D_c for the lattice expansion, please also give the associated standard uncertainty (standard deviation).*

The values have been given now.

2. *On page 10 line 325, there are 4 periods at the end of the line.*

Thanks – this has been fixed.

3. *In Figure 8e, the authors give microstructure analysis and list microstrain fluctuations under a heading “macrostrain fluctuation”; please correct to “microstrain fluctuations”*

This has been fixed.